# Quantum dot spin coherence governed by a strained nuclear environment

R. Stockill[1], C. Le Gall[1], C. Matthiesen[1,†], L. Huthmacher[1], E. Clarke[2], M. Hugues[3] & M. Atatüre[1]

The interaction between a confined electron and the nuclei of an optically active quantum dot provides a uniquely rich manifestation of the central spin problem. Coherent qubit control combines with an ultrafast spin–photon interface to make these confined spins attractive candidates for quantum optical networks. Reaching the full potential of spin coherence has been hindered by the lack of knowledge of the key irreversible environment dynamics. Through all-optical Hahn echo decoupling we now recover the intrinsic coherence time set by the interaction with the inhomogeneously strained nuclear bath. The high-frequency nuclear dynamics are directly imprinted on the electron spin coherence, resulting in a dramatic jump of coherence times from few tens of nanoseconds to the microsecond regime between 2 and 3 T magnetic field and an exponential decay of coherence at high fields. These results reveal spin coherence can be improved by applying large magnetic fields and reducing strain inhomogeneity.

[1] Cavendish Laboratory, University of Cambridge, JJ Thomson Avenue, Cambridge CB3 0HE, UK. [2] EPSRC National Centre for III-V Technologies, University of Sheffield, Sheffield, S1 3JD, UK. [3] CNRS-CRHEA, rue Bernard Grégory, Valbonne 06560, France. † Present address: Department of Physics, University of California, Berkeley, California 94720, USA. Correspondence and requests for materials should be addressed to R.S. (email: rhjs2@cam.ac.uk) or to M.A. (email: ma424@cam.ac.uk).

Decoupling techniques, such as Hahn echo, protect quantum states by filtering the effects of correlated environment noise[1,2]. They are essential to realising quantum information, computation and simulation protocols by extending the usable coherence time of quantum states, but also provide a powerful spectroscopic tool to understand environmental dynamics[3,4]. A particularly rich environment is experienced by electrons confined to self-assembled indium–gallium–arsenide (InGaAs) quantum dots: the electron spin couples via the contact hyperfine interaction to the dense nuclear spin bath of the $\sim 10^4 - 10^5$ atoms which constitute the quantum dot. This nuclear spin bath is subject to inhomogeneous electric field gradients within the crystal lattice, which arise from the strain-driven self-assembly during epitaxial growth. Local field gradients are present in addition as a consequence of the random alloying of atomic species in these systems. The nuclear spins in the quantum dot couple to these gradients via quadrupolar moments, resulting in spatially-dependent shifts of their energy levels, which has recently been shown to provide a natural isolation from dipolar interactions[5]. The effect of the hyperfine interaction is a loss of electron spin coherence within a few nanoseconds in time-averaged measurements[6,7]. While Hahn echo decoupling has been shown to enable the recovery of spin coherence for up to $3\,\mu s$[8], identifying the irreversible mechanism preventing coherence from reaching closer to the millisecond spin relaxation time[9,10] is an unresolved issue.

Recent studies of electron-spin dynamics have provided a comprehensive understanding at zero and very low (mT) magnetic field[11,12], but dephasing mechanisms and the limits of spin coherence in self-assembled quantum dots remain unclear at larger fields (few T) where quantum-dot spins are promising quantum bit candidates. A key challenge to understand electron-nuclear dynamics at non-zero external magnetic fields in these structures is presented by dynamic nuclear spin polarization. This nonlinear measurement-induced process, often observed as the dragging of resonance frequencies by detuning resonant optical probes[13], affects both the evolution of electron spin states and the detuning of resonant spin readout to the extent that it can limit state retrieval and obscure the mechanisms influencing the electron-spin state[14].

Here, we present an all-optical solution to access the quantum dot spin state dynamics free from polarizing the nuclear environment. We then use this method to implement Hahn echo decoupling[1] without perturbing the nuclear bath, and find the coherence to be governed by the quadrupolar coupling of the nuclear bath to the inhomogeneous strain fields in the quantum dot. In this way we recover the intrinsic limit to coherence for the central spin, indicating that extension beyond this few-microsecond bound necessitates the reduction of strain-induced quadrupolar broadening in these materials.

## Results

**Coherent electron spin control and electron-nuclear dynamics.** InGaAs quantum dots possess strong, coherent optical transitions[15,16], forming an ultrafast interface between the Zeeman-split ground states and well-defined optical modes[17–19]. The level structure allowing optical access to the spin state of an electron resident in the InGaAs quantum dot is depicted in Fig. 1a. An in-plane magnetic field permits four equal-strength near-infrared transitions to the excited trion states, at $969\,nm$ in our case, and splits the electron-spin ground states by $\Delta_e$, while the excited states are split by $\Delta_h$. Resonant addressing of an optical transition provides spin-dependent optical readout and prepares a well-defined ground state by optical pumping[20]. Application of a circularly polarized and spectrally detuned pulse rotates the

electron spin via the AC-Stark shift[21]. This can be observed in the fluorescence rate as power-dependent spin-Rabi oscillations, plotted in Fig. 1b, when the rotation pulses are accompanied by a resonant drive for spin readout and repump. In our experiments we achieve electron spin rotation with error rates of $\leq 3\%$ using 2 ps-long pulses.

An electron spin superposition stored in the quantum dot evolves at the spin splitting $\Delta_e$. This frequency, principally determined by the external quantization field $\mathbf{B}_{ext}$, is influenced by the contact hyperfine interaction with the nuclear spins within the electronic wavefunction. This interaction can be represented semi-classically by an effective magnetic field, the Overhauser field $\mathbf{B}_{OH}$, as depicted in Fig. 1c. In the limit of $B_{ext} \gg B_{OH}$, Overhauser field components parallel to the electron quantization couple linearly to the spin splitting, while perpendicular terms perturb the splitting quadratically[22,23]. The high-frequency dynamics of the Overhauser field are determined by the nuclear bath precessing in the external field, together with the coupling to

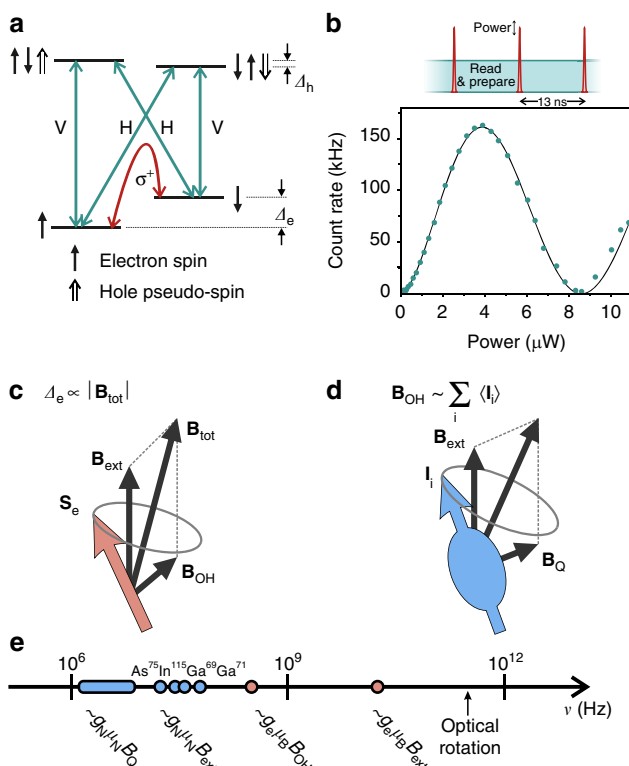

**Figure 1 | Optical access to InGaAs quantum dot spins and relevant spin dynamics. (a)** Energy level structure for a negatively charged quantum dot in Voigt geometry, showing the optically allowed transitions and their respective polarizations. The red arrow represents the effective coherent coupling between the spin ground states, which we create using a $\sigma^+$—polarized picosecond pulse detuned by $\sim 3\,nm$ from the optical resonance. **(b)** Pulse sequence and recorded count rates for the control of a single electron spin. The depicted scheme contains both the picosecond rotation pulse and the continuous spin readout. The curve in the readout count rate is a sinusoidal fit with a sublinear power dependence. **(c)** The electron spin splitting is due to both the external magnetic field and the hyperfine coupling with the nuclear bath, represented by the Overhauser field $\mathbf{B}_{OH}$. **(d)** The Overhauser field is set by the average spin projection $\sim \Sigma_i \langle \mathbf{I}_i \rangle$, where $\mathbf{I}_i$ is the spin at each atomic site. The nuclear spins are subject to the external field and quadrupolar coupling to electric field gradients, represented here by the effective field $\mathbf{B}_Q$. **(e)** The frequency axis displays the relevant frequencies for nuclear and electron dynamics at the few-Tesla external field regime, as well as the optically induced electron spin Rabi frequency.

nuclear site-dependent electric field gradients via quadrupolar moments, represented in Fig. 1d for an individual nuclear spin ($I_i$) as an effective field $\mathbf{B}_Q$[24]. The sum effect of the two is such that the resulting Overhauser field fluctuations contain components both parallel and perpendicular to the external field direction. The frequency chart in Fig. 1e outlines the relevant timescales for each of these processes in a few-Tesla external field. At these fields, the fastest process is electron spin precession at $g_e\mu_e \sim 6\,\mathrm{GHz\,T^{-1}}$, broadened by the Overhauser field width $g_e\mu_e B_{OH} \sim 100\,\mathrm{MHz}$[25,26]. Atomic-species dependent nuclear Zeeman splitting of $g_N\mu_N \sim 10\,\mathrm{MHz\,T^{-1}}$ is supplemented by the quadrupolar coupling $g_N\mu_N B_{OH} \sim 1\text{–}10\,\mathrm{MHz}$[27]. The effect of other dynamics, such as dipolar interaction between nuclear spins[5], electron-mediated nuclear flip flops[28] or the precession of nuclei due to hyperfine coupling with the electron[29,30] can be neglected in this regime due to their weaker coupling strengths. For reference, we include the Rabi frequency of our spin control at $\sim 500\,\mathrm{GHz}$, which is at least an order of magnitude faster than any other process.

**Suppressing dynamic nuclear spin polarization.** The nuclear dynamics introduced in Fig. 1d affect the storage and retrieval of arbitrary spin states from the quantum dot. To investigate the extent of their effect we measure the Ramsey interference of spin states for varying delay, $\tau$ between two $\pi/2$ spin rotations. The sub-panels of Fig. 2a, labelled (i–iv), display the resonant spin readout signal when we probe each of the four available transitions (see level schemes). All data sets show a striking departure from the expected sinusoidal beating associated with the precession of the electron spin. When probing the spin-down projection (iii & iv), a hysteretic saw-tooth behaviour emerges[14], while for probing spin-up projection (i & ii) the readout signal is suppressed for delays beyond 200 ps[17]. This behaviour emerges from the dynamic polarization of the nuclear spins due to resonant driving of the optical transitions, producing a feedback loop between bath polarization and the measured electron state[13,31]. As can be seen in Fig. 2a, it is the ground-state electron projection of the readout transition which dictates the feedback dynamics we observe, consistent with mechanisms

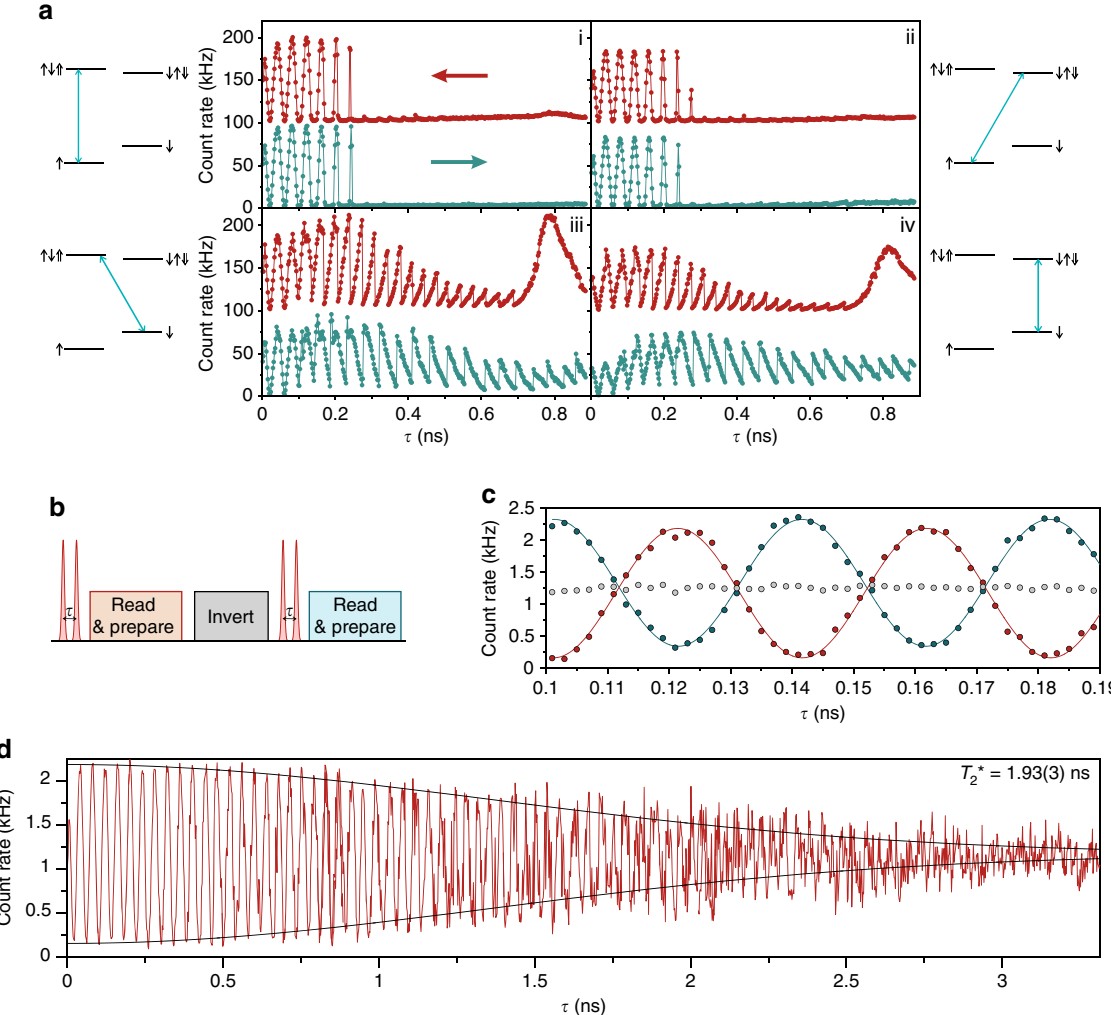

**Figure 2 | Nuclear polarization in Ramsey interferometry.** (**a**) (i–iv) Readout fluorescence as a function of the delay between $\pi/2$ rotations using the four allowed transitions to probe and prepare the electron spin, and for different scan directions. The level schemes indicate the probed transition. The non-sinusoidal shape is due to the polarization of nuclear spins in the quantum dot. The data are taken for $B_{ext} = 4\,\mathrm{T}$ resulting in an electron spin splitting of 25.2 GHz. (**b**) Alternating pulse sequence to suppress nuclear spin polarization. Every second rotation sequence begins with an inverted spin state. (**c**) Count rates from the alternating sequence. The two out-of-phase signals with (blue circles) and without (red circles) a spin inversion produce a time-averaged signal without phase dependence (grey circles). The curves are sinusoidal fits to data. The lower count rates compared with (**a**) are a consequence of the hardware constraints in producing and measuring the alternating sequence. (**d**) Full free induction decay with alternating sequence. The probed transition is the same as in **a** (i). The black curve is a Gaussian envelope with a 1.93-ns decay.

based on the non-collinear hyperfine interaction[32]. The effect of this state-dependent polarization is to frustrate time-averaged measurements of the electron-nuclear system and to prevent access to the full timescale of spin coherence. To inhibit the polarization of the nuclear bath we repeat the same measurement as in Fig. 2a, however we invert the spin before every other repeat, as depicted in Fig. 2b. The two resulting measurements presented in Fig. 2c suppress the phase dependence in the average signal (grey circles). In this way we prevent the build-up of nuclear polarization and resolve the evolution of the unperturbed electron-nuclear system, as shown in Fig. 2d for one half of the measurement pair. The symmetric, Gaussian-envelope decays with $T_2^{HE} = 1.93(3)$ ns which is testament to the large amplitude, yet quasi-static environmental noise, consistent with an Overhauser field standard deviation of 33 mT[11,30] (Supplementary Fig. 1 and Supplementary Note 1).

**Nuclear bath evolution dominating electron spin coherence.** Having successfully decoupled nuclear bath polarization from our resonant optical spin readout, we implement a Hahn echo scheme to assess the extent to which we can protect electron-spin coherence. The measurement sequence is displayed in Fig. 3a for a 91.2-ns delay between the $\pi/2$ rotations, in which the central $\pi$ rotation refocuses the dephased spin state. We sweep the central rotation by $\tau < T_2^*$ and record the visibility of the oscillatory readout signal[8]. In the limit of perfect rotations the visibility is a direct measure of the spin-state coherence. Figure 3b displays the integrated readout from the pulse sequence in Fig. 3a. The two traces correspond to the echo signal with (blue circles) and without (red circles) initial inversions. As with the free-induction decay in Fig. 2, decoupling the readout from bath polarization is essential to obtain a meaningful readout signal (Supplementary Fig. 2 and Supplementary Note 2).

Figure 3c shows the visibilities we recover when varying storage time, $T$, up to 1.3 μs for four external magnetic fields. The refocusing $\pi$ pulse filters the quasi-static noise components that resulted in the 2-ns time-averaged dephasing, yet the electron state is still susceptible to high-frequency dynamics of the Overhauser field. The data reveal a marked, nonlinear dependence on the external magnetic field strength: at low fields (2 T, top panel) spin coherence decays almost completely within 20 ns, followed by a very weak revival after ∼125 ns. At higher magnetic fields coherence times are dramatically prolonged, featuring damped oscillatory behaviour at short times and an exponential tail for longer delays.

The quality of the experimental data enables us to employ the Hahn echo as a sensitive spectroscopic probe of the nuclear bath dynamics: in protecting the electron spin coherence it provides a tunable filter of the Overhauser field noise spectrum. We can fully explain the measured spin coherence by considering the hyperfine interaction with an ensemble of non-interacting Ga[69], Ga[71], As[75] and In[115] nuclear spins in the presence of an external magnetic field and a significant quadrupolar interaction. A calculation of the nuclear noise spectra, presented in Fig. 3d, allows us to simulate the Hahn echo decay function (grey curves in Fig. 3c) which are in good quantitative agreement with the data. In general, we find the partial collapse and revival of the spin-echo curve to be due to linearly coupled Overhauser field components along the electron-spin quantization direction. The second-order coupling to the perpendicular Overhauser components produces a smooth envelope function responsible for the exponential tail observed at higher magnetic fields.

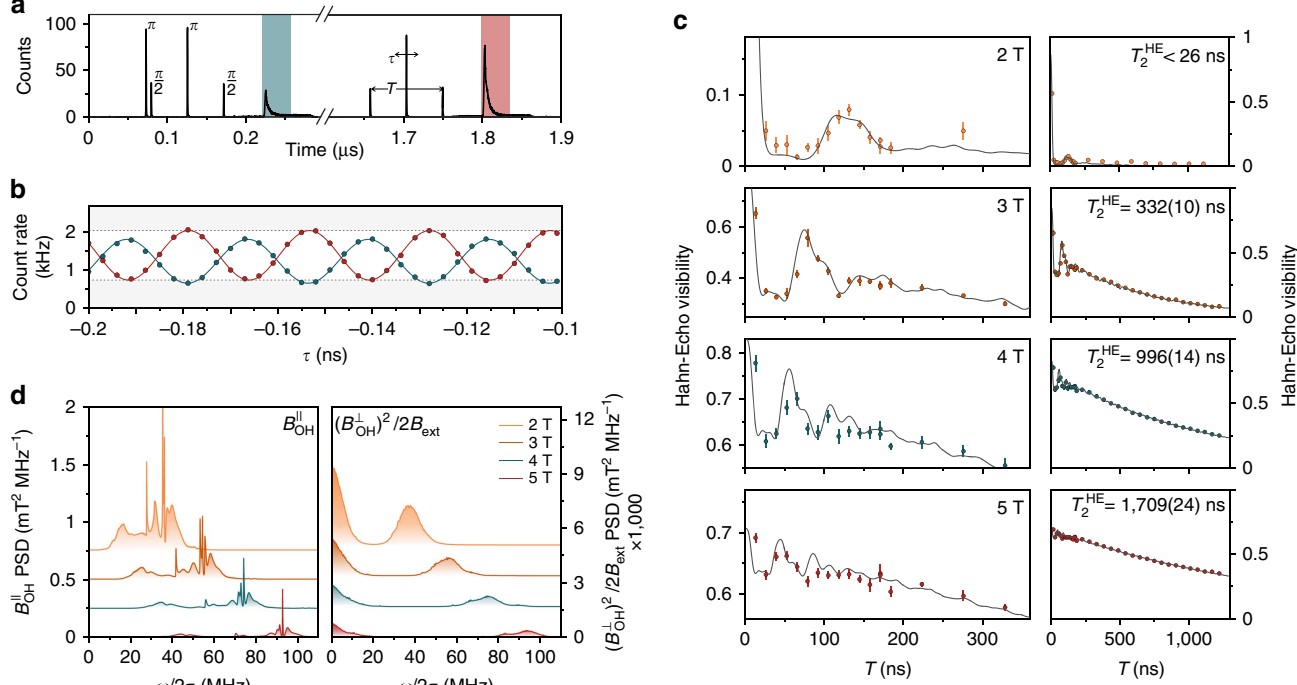

**Figure 3 | Hahn echo measurement with suppressed nuclear feedback.** (**a**) Histogram of pulse sequence to perform and measure all-optical Hahn echo for a storage time, $T$ of 91.2 ns. The marked areas are regions of interest for spin readout. (**b**) Integrated count rates in regions of interest for the two readout pulses as the central rotation pulse is scanned by $\tau$, at an external field of 3T, revealing a visibility of 47.6(1.2)%. The blue trace is of smaller visibility due to imperfect spin inversion. (**c**) Extracted visibilities for varying storage time, $T$, and external field values. The visibilities are drawn from the readout without initial inversion. Error bars show the s.e.m. for repeated measurements. Right-hand panels show zoomed-out traces of long storage times, showing the exponential visibility decay and the fitted decay times. The solid curves in all panels are the result of modelling the echo sequence as a spectral filter of the nuclear spectra in (**d**). (**d**) Power spectral densities of the nuclear noise. The left panel displays the linearly coupled Overhauser field components, along the quantization direction of the electron. The right panel displays the spectra of quadratically coupled transverse components.

The left panel of Fig. 3d shows the linearly coupled nuclear noise spectra while the right panel shows the transverse quadratically coupled counterpart. The broad features result from integrating over a Gaussian distribution of quadrupolar energies and orientations corresponding to the inhomogeneous spread of electric field gradients (Supplementary Note 3). Our choice of physical parameters (Supplementary Tables 1 and 2) is based on an indium concentration of 0.5 with strain profiles motivated by atomistic calculations[33,34]. Fitting the power spectrum amplitude to our echo modulation is consistent with an Overhauser field s.d. of 28 and 40 mT for the quadratically and linearly coupled components, respectively. By resolving the nuclear spin spectra into the constituent atomic species we find that indium plays the dominant role in the dephasing of the electron, owing to its large nuclear spin, while the gallium isotopes have a negligible effect (Supplementary Fig. 3).

The striking magnetic field dependence can be understood by considering the competition between the nuclear Zeeman and the quadrupolar interaction. The large strain inherent to InGaAs quantum dots results in a quadrupolar coupling strength equivalent to a $\sim 1$-T magnetic field. As depicted in Fig. 1d, a tilt between the effective quadrupolar field axis and the external field leads to parallel Overhauser field components in the hyperfine interaction, which dominate at low external magnetic fields. The strong inhomogeneous broadening of the quadrupolar coupling prevents any meaningful rephasing of this large amplitude term below a 2-T magnetic field. Electron spin coherence is then effectively suppressed beyond $\sim 20$ ns. At higher magnetic fields the Zeeman interaction starts to dominate nuclear dynamics, reducing the amplitude of linearly coupled Overhauser field components (Supplementary Fig. 4). Consequently, we observe a strong increase of spin coherence between external field values of 2 and 3 T. At higher fields the oscillation amplitude in the Hahn echo signal dies away and spin coherence decays exponentially at a field-dependent rate extending to 1.7 µs at 5 T. This decay stems from the difference frequency terms in the second-order hyperfine coupling, which result in a broad low-frequency shoulder, plotted in the right panel of Fig. 3d. The continuous spread of noise components between 0 and $\sim 10$ MHz causes a loss of spin coherence without revival. In this way, the quadrupolar inhomogeneity bounds the electron spin coherence to this microsecond timescale, set by the square of the Overhauser field strength. These components couple to the electron splitting with a $1/B_{ext}$ dependence, which reduces the effective amplitude at higher fields. The data in Fig. 3c, and further data taken for a different quantum dot at a 7 T external field yielding $T_2^{HE} = 2.7 \mu s$ (Supplementary Fig. 5), support this expectation.

Calculations of a detailed magnetic field dependence of the Hahn echo decay based on the nuclear bath parameters found from fitting the experimental data are presented in Fig. 4, which summarizes our results: A clear transition from suppressed spin coherence with dephasing times of 10–20 ns to microsecond spin coherence takes place between 2 and 3 T, highlighted in the figure sub-panel. Following this transition electron spin coherence increases approximately linearly with magnetic field at a rate of $\sim 0.7 \mu s\,T^{-1}$, exceeding 5 µs at 10 T. We note this linear trend is in agreement with earlier results at intermediate magnetic fields[8], but differs for high fields (Supplementary Note 3).

## Discussion

By inhibiting dynamic nuclear spin polarization during resonant optical access to spin states, we observe the quadrupolar interaction of the nuclear bath with inhomogeneous electric field gradients providing an intrinsic bound to spin coherence in InGaAs

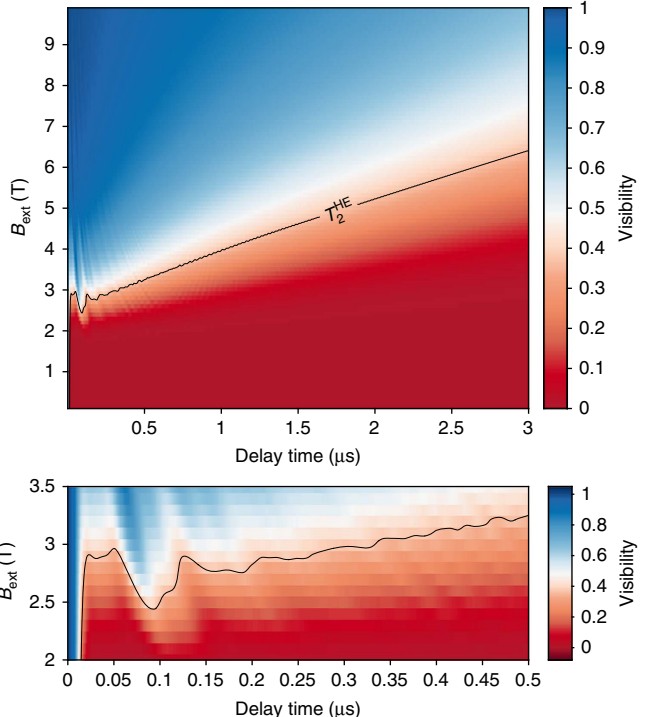

**Figure 4 | Magnetic field dependence of Hahn echo decay.** Calculated Hahn echo visibility as a function of delay time and external magnetic field for the same parameters used in Fig. 3. The degree of visibility is colour-coded according to the scale to the right of the figure. The contour highlights where the visibility drops to $1/e$ as a function of the two parameters. The sub-panel contains a zoom on the 2–3.5 T region, featuring the dramatic rise in spin coherence.

quantum dots. In other systems, such as electrostatically defined GaAs quantum dots, the nitrogen-vacancy centre in diamond or phosphorus donors in silicon, multi-pulse decoupling schemes can be used to prolong the spin coherence beyond the Hahn echo limits[23,35,36]. Such schemes rely on noise being correlated on timescales longer than the pulse separation[7]. The broadband quadrupole-coupled nuclear dynamics in our self-assembled quantum dots present a corresponding timescale of $\sim 50$ ns and only pulses spaced more closely than this could extend spin coherence beyond the times we observe. In agreement, at low-magnetic fields we find we can extend coherence for pulse separations of 6.5 ns, where the Hahn echo limit of $\sim 20$ ns can be pushed beyond 65 ns using 9 $\pi$ pulses (Supplementary Fig. 6). Further decoupling is limited in our case by pulse imperfections. While these imperfections may be improved using composite pulses[37], the number of rotations possible will be limited by irreversible excitonic dephasing through coupling to acoustic phonons[38,39], such that an improvement over Hahn echo $T_2$ times in the high-magnetic field regime would face challenges.

The coherence of electron spins in electrostatically defined GaAs quantum dots is limited by quadrupolar nuclear broadening under certain conditions[40], but the combination of a larger nuclear bath, the absence of inherent strain, and the lack of indium has allowed retention of coherence for times close to 1 ms[41]. By suppressing inter-nuclear dipole–dipole coupling, the strain-broadening in self-assembled quantum dots has been shown to preserve nuclear coherence beyond these unstrained systems[5,42], while however preventing rephasing of the Overhauser field.

A single heavy-hole as a quantum-dot spin qubit should allow for longer dephasing times due to a much weaker hyperfine

interaction, although other mechanisms can limit its coherence[43,44]. Alternatively, longer coherence times in self-assembled quantum dots could be realised either for systems without quadrupolar nuclear moments, as in II–VI quantum dots[45], or through strain-free growth methods[46,47]. One particular example of a strain-free system is provided by optically active GaAs/AlGaAs quantum dots grown by droplet epitaxy. With over two orders of magnitude weaker local strain than the quantum dots used here, significantly different nuclear dynamics have been reported[48]. With reduced quadrupolar broadening absolute coherence times that are competitive with other spin qubit realizations could be achieved, while still maintaining the ultrafast control rates and optical integration the material properties enable.

## Methods

**Sample.** We use a sample grown by molecular beam epitaxy containing a single layer of self-assembled InGaAs quantum dots in a GaAs matrix, embedded in a Schottky diode for charge state control. The Schottky diode structure comprises an $n^+$-doped layer 35 nm below the quantum dots and a ∼6-nm thick partially transparent titanium layer evaporated on top of the sample surface. This device structure allows for deterministic charging of the quantum dots and shifting of the exciton energy levels via the DC Stark effect. Twenty pairs of AlAs/AlGaAs layers form a distributed Bragg reflector below the quantum dot layer for increased collection efficiency in the spectral region between 960 and 980 nm. Spatial resolution and collection efficiency are enhanced by a zirconia solid immersion lens in Weierstrass geometry positioned on the top surface of the device. The device is cooled in a liquid-helium bath cryostat to 4.2 K and surrounded by a superconducting magnet.

**Spin inversion to prevent nuclear spin polarization.** The inversion necessary to cancel phase terms in the average readout signal and suppress nuclear polarization can be provided by either a coherent π-rotation, or an incoherent re-pumping step. For Hahn echo the short spin evolution measured (∼200 ps) is such that an imperfect π-rotation suffices; however for measurements of the time-averaged dephasing, the longer delays measured between the π/2-rotations and the corresponding enhanced sensitivity to nuclear back-action require complete spin inversion to cancel the time-averaged phase. Due to this we supplement our imperfect spin rotation with a pumping step.

**Pulse sequence and detection.** Optical pulse sequences are constructed from a Ti:Sapphire pulsed laser in picosecond-mode, detuned from the optical resonance by 3 nm and a resonant continuous-wave diode laser. Both are modulated with fibre-coupled waveguide electro-optic modulators. The modulators are locked to the 76-MHz repetition rate of the pulsed laser via a pulse delay generator with 8-ps jitter. Additional suppression of the readout pulse is provided by an acousto-optical modulator, realising 6,000:1 suppression of readout lasers and >2,000:1 rotation pulse suppression. The readout laser is used at a power below optical saturation to avoid spin-pumping when not reading the spin state. The readout fluorescence is filtered from the resonant laser by polarization mode rejection. Additional filtering of the detuned rotation pulses is provided by a holographic grating with a 30-GHz full width at half maximum and the first-order diffraction efficiency above 90%. The photon detection events are recorded with a time-correlation unit and a single photon detector with a timing resolution of 350 ps.

**Data availability.** All relevant data are available from the authors on request.

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

## Acknowledgements

We gratefully acknowledge financial support by the University of Cambridge and the European Research Council ERC Consolidator grant agreement no. 617985. C.M. acknowledges Clare College, Cambridge, for financial support through a Junior Research Fellowship. We thank H. Bluhm, T. Botzem, L. Cywinski, D. Kara, M. Stanley and J.M. Taylor for fruitful discussions and S. Topliss for technical assistance.

## Author contributions

R.S., C.L.G., C.M., L.H. and M.A. devised and designed the experiments. M.H. and E.C. grew the sample. C.M. processed the devices. R.S., C.L.G., C.M. and L.H. performed the measurements and analysed the data. C.M. and C.L.G. performed the theoretical modelling shown in Fig. 3. All authors contributed to the discussion of results and preparation of the manuscript.

## Additional information

**Competing financial interests:** The authors declare no competing financial interests.

