## [Peer review file · Nature Communications]

Reviewers' Comments:

Reviewer #1 (Remarks to the Author)

This paper presents an optical study of the role of nuclear spin in the decay of spin coherence of an electron in an InGaAs quantum dot. Suppression of measurement-induced nuclear field effects are nicely suppressed, leading to a measurement of the intrinsic effects of the nuclear spin on the electron spin dephasing. Theoretical modeling provides a detailed interpretation of the dynamics, and in particular, the important role of strain inhomogeneity and quadrupolar interactions.

The work is impressive and convincing. The experiment and theory are of high quality. The paper is clear and well presented. This research area of spin relaxation in quantum dots continues to be heavily studied both because of the interesting physics of the central spin problem and because of its potential in quantum information technology. Optical techniques used in this study as well as the conclusions on the role and limitations of nuclear spin in electron spin dephasing in strained quantum dots will be of high interest to those in this research field. I think that this paper should essentially be published as is.

The only error that I noticed is a typo in Fig. 1 and its caption: section 1d, which is referenced in the text, seems to be missing in the figure.

Reviewer #2 (Remarks to the Author)

Stockill and co-authors report a comprehensive study of InGaAs quantum dot (SAQD) spin coherence under the strained environment typical of self-assembly. They use all-optical methods to initialize, rotate, read, and sustain the spin states. By inverting the initialized spin before every control cycle, the build up of nuclear spin polarisation is prevented. The authors then implement a Hahn-echo experiment to study the quantum state coherence on microsecond timescales under the influence of an Overhauser field and external magnetic field. They show an excellent match between experimentally measured visibility decay curves and calculated results based on the hyperfine interaction with nuclear spins. Combining the experimental data and calculations, the authors quantitatively analyze how the nuclear spin bath in the strained environment serves as the dominant limitation of sustaining the coherence in SAQDs.

Overall, this high-quality research is presented very well in this manuscript. Although similar experiments have been conducted previously (Ref. 8 in the manuscript), this manuscript focuses on the factors related to the nuclear bath environment. A systematic experiment with nearly perfect agreement between the experiment and theoretical model suggests that the authors have excluded other related factors and gained a much deeper understanding of the origin of this striking result. Because maintaining spin coherence in SAQDs is an important topic, this piece of work could be inspiring and instructive to future research. The results and conclusions also seem reliable to me. I recommend publishing this manuscript with a few minor revisions.

1. Fig 1c: the authors should specify what the terms on the scheme, e. g., Ii, denote. I also suggest the authors mark the linearly- and quadratically-coupled Overhauser components for clarity.

2. Line 95 of the main text: Could the authors give an explanation about the different feedback dynamics for probing emissions with spin-up and spin-down ground states? If it is less related to

the topic, the authors could put this part of the content in the supplementary material.

3. Fig. 3 (c): When the authors mention "the curve" as the modeling results, does that mean the solid lines shown in both the left and right panels?

4. Fig. 4: This figure shows the Hahn-echo visibility as a function of delay time and external magnetic field. It exhibits the trend in a high magnetic field very clearly but lacks details in the low magnetic field/ low delay time region. I suggest the authors add another figure to show a zoom-in on the range of a few hundreds of nanoseconds so we will be able to see the drastic increase of the Hahn-echo decay time before B_{ext} reaches 3 T.

5. Supplementary, page 13: The authors point out that the linear increase of Hahn-echo coherence with increasing magnetic field is different from the results reported by Press et al. in Ref. [8] and give a discussion about possible reasons. The authors should mention this disagreement in the main text and refer to the supplementary.

Reviewer #3 (Remarks to the Author)

In their manuscript, the authors present ingenious experiments and careful theoretical analysis on the coherence of a single spin in a self-assembled InGaAs quantum dot under the external magnetic field of a few tesla. They together convincingly reveal that the spin coherence time of this system is limited by the strain-induced inhomogeneous quadrupolar fields, which have so far been masked by the (notorious) dynamic nuclear polarization (DNP) process. The authors suppress the buildup of DNP with alternating spin inversion, and the Hahn echo decay curves thus obtained are reproduced, with surprising accuracy, by a model considering both the Overhauser and quadrupolar fields

I recommend this manuscript for publication in Nature Communications because of its scientifically sound and high quality data (both experiment and theory) as well as the importance in the corresponding research fields. The present work provides a clear solution and a simple physical picture to the unsolved "central spin problem" in the system of InGaAs-based self-assembled quantum dots. This work will also call for serious reconsideration of materials to be used, and I am looking forward to the future development in this direction.

I have a few questions regarding some details. Otherwise I am satisfied with this carefully prepared manuscript.

(i) In figure 2(a), the maximum count rate is around 100 kHz, whereas in figure 2(b) it drops down to a few kHz, contrary to the naïve expectation of 50 kHz. How long does the spin inversion in this Ramsey fringe experiment take? Or are there other reasons which significantly reduce the measurement efficiency?

(ii) The electron Zeeman splitting is 25.2 GHz at $B_{\text{ext}} = 4$ T, so the g-factor seems to be about 0.45. This value is close to the one obtained by Press et al. (0.442, reference [8]) and somewhat deviates from the one by Bechtold et al. (0.55, reference [11]). Naively, these values (and also the emission wavelengths without the Stark effect) to some extent reflect the Indium contents in quantum dots. On the other hand, T_2 obtained by these groups are 996 ns (the present work), ~ 2.5 μs (Press et al.), and ~ 1.3 μs (Bechtold et al.) at 4 T, which do not correlate with the g-factors (and possibly with the Indium contents). Are these differences simply the dot-to-dot variations? Can the authors' calculation give these differences if some parameters are tweaked? In particular, Bechtold et al. write in page 4 of reference [11] that "Our theoretical model based only on quadrupole and hyperfine interactions do not explain this relaxation at large fields", referring to the mono-exponential Hahn echo decay. Please comment on these issues.

(iii) To me, the axes of figures 4 may better be inverted, i.e., the horizontal axis should be the magnetic field, so that the magnetic field dependence of T_2 is clearer. Or a separate figure showing the calculations of B_{ext} vs. T_2 for various Indium contents (also plotting the

experimental points corresponding to figure 3 and figure S3 for the $x = 0.5$ case) is beneficial to the readers, if the authors' calculation has enough predictive power. In particular, the calculation on strain-free GaAs-based quantum dots ($x = 0$), which the authors expect promising, is intriguing. The calculation on different Indium contents may also answer my concerns in (ii).

Finally, below is a list of minor typographical errors I have noticed.

- (1) In line 11 (the number in the left-hand side of the manuscript) of page 2, "Hahn-echo" should read "Hahn echo". In addition, I personally feel that the hyphenation is unnecessary for "Hahn-echo decoupling", "Hahn-echo scheme", and so on. These are unmistakable.
- (2) In line 69 of page 3, line 84 of page 4, and line 158 of page 6, the referred "Fig. 1(d)" does not exist. It actually refers to the right-hand side of Fig. 1(c).
- (3) In reference [8], the eighth author should be "S. Höfling".
- (4) In reference [11], the seventh author should be "K. Müller".
- (5) In reference [15], the article number should be "093602".
- (6) In reference [18], the eleventh author should be "S. Höfling".
- (7) In reference [19], the fourth and seventh authors should be "L.-M. Duan" and "A. S. Bracker", respectively.
- (8) In reference [20], the seventh author should be "A. S. Bracker", and the article number should be "097401".
- (9) In reference [24], the third author should be "S. A. Crooker".
- (10) In reference [25], the seventh and eighth authors should be "V. K. Kalevich" and "K. V. Kavokin", respectively.
- (11) In reference [27], the fourth, sixth, seventh and tenth authors should be "A. B. Krysa", "A. D. Andreev", "A. M. Sanchez", and "A. I. Tartakovskii", respectively.
- (12) In reference [43], the sixth author should be "A. Imamoğlu", and the publication year should be "2016".

Reviewer #1 (Remarks to the Author):

This paper presents an optical study of the role of nuclear spin in the decay of spin coherence of an electron in an InGaAs quantum dot. Suppression of measurement-induced nuclear field effects are nicely suppressed, leading to a measurement of the intrinsic effects of the nuclear spin on the electron spin dephasing. Theoretical modeling provides a detailed interpretation of the dynamics, and in particular, the important role of strain inhomogeneity and quadrupolar interactions.

The work is impressive and convincing. The experiment and theory are of high quality. The paper is clear and well presented. This research area of spin relaxation in quantum dots continues to be heavily studied both because of the interesting physics of the central spin problem and because of its potential in quantum information technology. Optical techniques used in this study as well as the conclusions on the role and limitations of nuclear spin in electron spin dephasing in strained quantum dots will be of high interest to those in this research field. I think that this paper should essentially be published as is.

The only error that I noticed is a typo in Fig. 1 and its caption: section 1d, which is referenced in the text, seems to be missing in the figure.

We thank reviewer #1 for pointing out the missing figure label. This has been corrected. We have further added a label for the frequency chart to avoid any confusion.

Reviewer #2 (Remarks to the Author):

Stockill and co-authors report a comprehensive study of InGaAs quantum dot (SAQD) spin coherence under the strained environment typical of self-assembly. They use all-optical methods to initialize, rotate, read, and sustain the spin states. By inverting the initialized spin before every control cycle, the build up of nuclear spin polarisation is prevented. The authors then implement a Hahn-echo experiment to study the quantum state coherence on microsecond timescales under the influence of an Overhauser field and external magnetic field. They show an excellent match between experimentally measured visibility decay curves and calculated results based on the hyperfine interaction with nuclear spins. Combining the experimental data and calculations, the authors quantitatively analyze how the nuclear spin bath in the strained environment serves as the dominant limitation of sustaining the coherence in SAQDs.

Overall, this high-quality research is presented very well in this manuscript. Although similar experiments have been conducted previously (Ref. 8 in the manuscript), this manuscript focuses on the factors related to the nuclear bath environment. A systematic experiment with nearly perfect

agreement between the experiment and theoretical model suggests that the authors have excluded other related factors and gained a much deeper understanding of the origin of this striking result. Because maintaining spin coherence in SAQDs is an important topic, this piece of work could be inspiring and instructive to future research. The results and conclusions also seem reliable to me. I recommend publishing this manuscript with a few minor revisions.

1. Fig 1c: the authors should specify what the terms on the scheme, e. g., l_i , denote. I also suggest the authors mark the linearly- and quadratically-coupled Overhauser components for clarity.

We thank the reviewer for bringing this to our attention. We have now specified in the figure caption what the terms denote. We tried adding an additional set of axes to the pictorial representations of the electron splitting to include the linear and quadratic projections of the Overhauser field. However, we feel this made the figure very hard to read. An illustration of this projection is given in Ref. 23 already, which is cited in the figure description in the main text.

2. Line 95 of the main text: Could the authors give an explanation about the different feedback dynamics for probing emissions with spin-up and spin-down ground states? If it is less related to the topic, the authors could put this part of the content in the supplementary material.

A full investigation and modelling of the feedback dynamics was not the primary aim of this work, as they are a specific product of the repeated probing of the spin state under Ramsey interferometry. These dynamics are discussed for the specific case of driving the lower frequency transitions in reference 14 of the manuscript (PRL **105** 107401 (2010)), and for hole spins in reference 31 (PRB **89** 075316 (2014)), added in this revision. Details of the noncollinear interaction are provided in references 13 (Nat. Phys. **5** 758 (2009)) and 32 (PRL **108** 197403). A detailed theoretical and quantitative analysis of the feedback dynamics is beyond the scope of this particular work and will be discussed separately.

3. Fig. 3 (c): When the authors mention "the curve" as the modeling results, does that mean the solid lines shown in both the left and right panels?

This is correct. We have now replaced "curve" by "solid curves in all panels" in the figure caption to resolve the confusion.

4. Fig. 4: This figure shows the Hahn-echo visibility as a function of delay time and external magnetic field. It exhibits the trend in a high magnetic field very clearly but lacks details in the low magnetic field/ low delay time region. I suggest the authors add another figure to show a zoom-in on the range of a few hundreds of nanoseconds so we will be able to see the drastic increase of the Hahn-echo decay time before B_{ext} reaches 3 T.

We thank the referee for this suggestion. A second panel has been added to Fig. 4, showing the Hahn-echo visibility in the critical region of 2-3.5 T for time delays up to 500 ns.

5. Supplementary, page 13: The authors point out that the linear increase of Hahn-echo coherence with increasing magnetic field is different from the results reported by Press et al. in Ref. [8] and give a discussion about possible reasons. The authors should mention this disagreement in the main text and refer to the supplementary.

The referee's suggestion has been implemented in the main text. We also note that the discussion at the end of supplementary note 3 has been expanded following questions from reviewer #3.

Reviewer #3 (Remarks to the Author):

In their manuscript, the authors present ingenious experiments and careful theoretical analysis on the coherence of a single spin in a self-assembled InGaAs quantum dot under the external magnetic field of a few tesla. They together convincingly reveal that the spin coherence time of this system is limited by the strain-induced inhomogeneous quadrupolar fields, which have so far been masked by the (notorious) dynamic nuclear polarization (DNP) process. The authors suppress the buildup of DNP with alternating spin inversion, and the Hahn echo decay curves thus obtained are reproduced, with surprising accuracy, by a model considering both the Overhauser and quadrupolar fields

I recommend this manuscript for publication in Nature Communications because of its scientifically sound and high quality data (both experiment and theory) as well as the importance in the corresponding research fields. The present work provides a clear solution and a simple physical picture to the unsolved "central spin problem" in the system of InGaAs-based self-assembled quantum dots. This work will also call for serious reconsideration of materials to be used, and I am looking forward to the future development in this direction.

I have a few questions regarding some details. Otherwise I am satisfied with this carefully prepared manuscript.

(i) In figure 2(a), the maximum count rate is around 100 kHz, whereas in figure 2(b) it drops down to a few kHz, contrary to the naïve expectation of 50 kHz. How long does the spin inversion in this Ramsey fringe experiment take? Or are there other reasons which significantly reduce the measurement efficiency?

We thank the reviewer for this observation. The absolute count-rates in Figs. 2(a) and (c) are a direct consequence of the pulse sequence repetition rates (76 MHz for (a) and 2 MHz for (c)). The rate in the second case is limited by both the hardware dead-time in the creation of the alternating pulse sequence and the maximum trigger rate for the time-to-digital converter we use to separate the two out-of-phase state readouts. To address this issue, we have noted these details in the figure caption.

(ii) The electron Zeeman splitting is 25.2 GHz at $B_{\text{ext}} = 4$ T, so the g-factor seems to be about 0.45. This value is close to the one obtained by Press et al. (0.442, reference [8]) and somewhat deviates from the one by Bechtold et al. (0.55, reference [11]). Naively, these values (and also the emission wavelengths without the Stark effect) to some extent reflect the Indium contents in quantum dots. On the other hand, T_2 obtained by these groups are 996 ns (the present work), ~ 2.5 μs (Press et al.), and ~ 1.3 μs (Bechtold et al.) at 4 T, which do not correlate with the g-factors (and possibly with the Indium contents). Are these differences simply the dot-to-dot variations? Can the authors' calculation give these differences if some parameters are tweaked? In particular, Bechtold et al.

write in page 4 of reference [11] that "Our theoretical model based only on quadrupole and hyperfine interactions do not explain this relaxation at large fields", referring to the mono-exponential Hanh echo decay. Please comment on these issues.

The reviewer is mentioning a number of interesting, if complex, issues relating to how experimental results so far compare. We first need to point out that data treatment differs in these three works. Both Press et al. and Bechtold et al. use a normalised amplitude scale for the spin-echo results. In the case of Press et al., data are normalised to the fringe amplitude at a time delay of 132 ns (about 0.65 visibility), and no data for shorter time delays are available. Normalising our data to the 132-ns amplitude yields a 1/e decay time about 1.4 μ s. The normalisation procedure applied in Bechtold et al. is not clear.

This partly explains the large difference to the Press et al. results. Given the complexity of the nuclear spin bath it is difficult to make quantitative predictions for other quantum dot samples/growth protocols based on our data alone. This would either require clean experimental data of the spin echo modulations or nuclear bath spectra calculated from atomistic models, such as the ones of Bulutay. With this caveat in mind, we find that by reducing the indium content slightly ($x=0.4$), accompanied with the expected reduction in quadrupolar field strength and spread (following Bulutay, ref. 33), the transition to high spin coherence takes place at a magnetic field around 2 T and spin coherence at 4 T is predicted to be 1.5 μ s.

We find similar spin-echo behaviour for the few quantum dots measured at very similar wavelengths on our sample, but clearly further work is needed for a comparative analysis across different wafers, g-factors, and wavelengths.

To comment on the model in Bechtold et al.: It is the strong inhomogeneity of the quadrupole interaction which in second order of the hyperfine interaction (the perpendicular term of the Overhauser field) leads to the exponential decay observed in all measurements so far. A first-order treatment does not predict the exponential decay.

In order to reflect the reviewer's comments and questions we have extended the discussion at the end of supplementary note 3.

(iii) To me, the axes of figures 4 may better be inverted, i.e., the horizontal axis should be the magnetic field, so that the magnetic field dependence of T_2 is clearer. Or a separate figure showing the calculations of B_{ext} vs. T_2 for various Indium contents (also plotting the experimental points corresponding to figure 3 and figure S3 for the $x = 0.5$ case) is beneficial to the readers, if the authors' calculation has enough predictive power. In particular, the calculation on strain-free GaAs-based quantum dots ($x = 0$), which the authors expect promising, is intriguing. The calculation on different Indium contents may also answer my concerns in (ii).

We appreciate the reviewer's comment on the arrangement of Figure 4, however for consistency with the previous figures, in particular Fig. 3, we feel the x axis should remain as storage time. Concerning the T_2 dependence on indium content we hope the reply to point (ii) above answers the reviewer's question.

Finally, below is a list of minor typographical errors I have noticed.

(1) In line 11 (the number in the left-hand side of the manuscript) of page 2, "Hahn-echo" should read "Hahn echo". In addition, I personally feel that the hyphenation is unnecessary for "Hahn-echo decoupling", "Hahn-echo scheme", and so on. These are unmistakable.

(2) In line 69 of page 3, line 84 of page 4, and line 158 of page 6, the referred "Fig. 1(d)" does not exist. It actually refers to the right-hand side of Fig. 1(c).

(3) In reference [8], the eighth author should be "S. Höfling".

(4) In reference [11], the seventh author should be "K. Müller".

(5) In reference [15], the article number should be "093602".

(6) In reference [18], the eleventh author should be "S. Höfling".

(7) In reference [19], the fourth and seventh authors should be "L.-M. Duan" and "A. S. Bracker", respectively.

(8) In reference [20], the seventh author should be "A. S. Bracker", and the article number should be "097401".

(9) In reference [24], the third author should be "S. A. Crooker".

(10) In reference [25], the seventh and eighth authors should be "V. K. Kalevich" and "K. V. Kavokin", respectively.

(11) In reference [27], the fourth, sixth, seventh and tenth authors should be "A. B. Krysa", "A. D. Andreev", "A. M. Sanchez", and "A. I. Tartakovskii", respectively.

(12) In reference [43], the sixth author should be "A. Imamoğlu", and the publication year should be "2016".c

We thank the reviewer for their careful checking of the manuscript text and references. We have amended these errors in the manuscript.